# The Role of Aquaculture and Capture Fisheries in Meeting Food and Nutrition Security: Testing a Nutrition-Sensitive Pond Polyculture Intervention in Rural Zambia

**DOI:** 10.3390/foods11091334

**Published:** 2022-05-04

**Authors:** Alexander M. Kaminski, David C. Little, Lucinda Middleton, Muleya Syapwaya, Mary Lundeba, Jacob Johnson, Carl Huchzermeyer, Shakuntala H. Thilsted

**Affiliations:** 1Institute of Aquaculture, University of Stirling, Stirling FK9 4LA, UK; d.c.little@stir.ac.uk; 2WorldFish One CGIAR, Plot 18944 Lubansenshi, Lusaka 10101, Zambia; lulu.middleton1@gmail.com (L.M.); muleyasy@gmail.com (M.S.); m.lundeba@cgiar.org (M.L.); 3Research Institute for the Environment and Livelihoods, Charles Darwin University, Ellengowan Drive, Casuarina, NT 0810, Australia; 4Department of Ecosystem Science and Management, Penn State University, State College, PA 16801, USA; jjj5438@psu.edu; 5Deutsche Gesellschaft für Internationale Zusammenarbeit (GIZ), Kariba Road, Lusaka 10101, Zambia; carl.huchzermeyer@giz.de; 6WorldFish One CGIAR, Jalan Batu Maung, Bayan Lepas 11960, Malaysia; s.thilsted@cgiar.org

**Keywords:** aquaculture, fisheries, small-scale, nutrition-sensitive, food systems, polyculture, food and nutrition security, Lake Bangweulu, Zambia, Africa

## Abstract

This study tested the efficacy of a pond polyculture intervention with farming households in northern Zambia. Longitudinal data on fish consumption and the associated nutrient intake of households (*N* = 57) were collected over a six-month period (September 2019–March 2020). One group of people tested the intervention while another group that practiced monoculture tilapia farming, and a third group that did not practice aquaculture, acted as control groups. A similar quantity of fish was consumed on average; however, the associated nutrient intake differed, based on the quantity and type of species consumed, particularly for those who had access to pelagic small fish from capture fisheries. There was a decrease in fish consumption from December onward due to fisheries management restrictions. The ponds provided access to micronutrient-rich fish during this time. Pond polyculture can act as a complementary source of fish to capture fisheries that are subjected to seasonal controls, as well as to households that farm tilapia. Assessments of how aquatic foods can improve food and nutrition security often separate aquaculture and capture fisheries, failing to account for people who consume fish from diverse sources simultaneously. A nutrition-sensitive approach thus places food and nutrition security, and consumers, at the center of the analysis.

## 1. Introduction

There is a growing recognition that freshwater lakes and rivers in sub-Saharan Africa are crucial to the food and nutrition security of millions of people [1,2]. Pelagic small fish and wetland species are among some of the cheapest sources of animal foods and are seen as a lifeline for rural households that struggle to meet their food and nutrition needs [3]. Many of these fish are rich in essential long-chain polyunsaturated fatty acids (PUFAs), which are crucial for cognitive development in children during the first 1000 days of life [4,5]. The amount and frequency of consumption of individual species are often underrecognized, as they are frequently lumped into larger categories of “fish” or “seafood”. There are few records of the true extent of yields and distribution of freshwater fish species for human consumption in sub-Saharan Africa [6]. It is largely believed, however, that total yields in many of these capture fisheries are declining or stagnating, which, coupled with population growth, means that countries need to increasingly rely on other sources of fish to achieve food and nutrition security, either by importation or developing a domestic aquaculture industry [7,8]. The latter has long been touted as a solution to supplement fish supplies on the continent; however, yields are still far too small to mark significant shifts in consumption [9]. Aquaculture in sub-Saharan Africa is still mostly driven by large, commercial farms that supply expensive fish for high- and middle-income consumers in urban areas [10,11]. While there is some evidence that smallholder fish farmers manage to improve household food and nutrition security through the direct sale and/or consumption of fish, most farmers still struggle to produce fish (especially tilapia) successfully and consistently [12,13]. This is particularly the case for smallholder tilapia farmers in Zambia [14]. 

Aquaculture in Zambia is expanding and rapidly commercializing. The total production is made up exclusively of tilapia species. Certain indigenous tilapia species are farmed throughout the country, but most of the production is dominated by one non-native tilapia species (*Oreochromis niloticus*) [15]. Most of the fish consumed by Zambians come from freshwater capture fisheries, not from aquaculture, and are eaten as dried and/or smoked products [14]. Zambia has a high diversity of indigenous species available in markets throughout the year, constituting a critical animal-source food for most of the population [16,17]. Fish consumption is stratified along economic lines and poorer people tend to consume small, dried, cheap fish, while well-off people tend to consume large, fresh fish, such as farmed tilapia [18]. The potential of small indigenous fish species (SIS) is increasingly recognized as crucial to food and nutrition security in low- and middle-income countries, due to their superior micronutrient composition compared to common commercial species, such as tilapia [19,20]. Such perspectives emerged from studies in Bangladesh, where SIS contributed significantly to increases in micronutrient intake, particularly for pregnant and lactating women [21,22]. Greater benefits were realized when multiple species were produced in small homestead ponds, i.e., polyculture production (as opposed to single species in monoculture production) [23].

The principle of polyculture is to stock compatible fish species that occupy different trophic niches in a pond ecosystem, thereby utilizing the available resources more efficiently [24,25]. Such approaches usually consider sustainability issues, with the aim of improving production per unit per land/water and using less energy, resulting in lower food conversion ratios (FCR) and lower production costs [26]. In commercial systems, polyculture is implemented with the intended outcome to increase fish growth, achieve higher yields, and gain greater profitability [27]. In many extensive systems in rural areas, however, the unintentional entry of wild self-recruiting species is an outcome of the system itself, e.g., rice-field fisheries in Bangladesh. Such extensive polyculture systems have since been noted for their ecological and nutritional outcomes [28]. The systems provide many benefits, such as allowing for shorter production cycles, faster cash flows, and the intermittent harvesting of highly nutritious fish throughout the season, which do not need to be purchased and restocked from hatcheries [23]. This type of mixed-fish production is better suited for extensive systems that rely on natural rather than formulated feeds usually operated by poorer farmers as a means of livelihood [28]. 

In sub-Saharan Africa, few studies have incorporated SIS into polyculture systems, probably because, at face value, they offer little in the way of economic reward. One study did find that small fish generated more gross income because the biomass of small barbs was larger than tilapias in a pond [29], though this may speak more to the difficulties farmers face in rearing tilapia. There is very little commercial incentive to establish hatcheries for SIS, and due to their fragility, recruiting and stocking can be problematic [29]. The knowledge of the number and diversity of species suitable for aquaculture is, thus, extremely limited in the region. 

In many cases, however, SIS already exist in household ponds in small-scale systems, especially in northern Zambia [30]. This is largely an unintentional consequence of the design of extensive pond systems that allow fish to enter and breed in the pond. Most ponds are also dug in local wetlands where there is an abundance of SIS. The benefit is that farmers can bypass the issue of procuring species from hatcheries or recruiting stock from larger capture fisheries. Smallholder farmers in northern Zambia, therefore, operate de facto polyculture systems. This fact is frequently unacknowledged in assessments of extensive, small-scale aquaculture systems in the region. Farmers are, however, actively encouraged by the government and development organizations to establish monoculture systems with local tilapias purchased from government hatcheries (there are almost no private hatcheries) to maximize the potential growth of single species for markets. As was the case in Bangladesh in the past, the SIS are treated as competitors with tilapia for pond resources. Farmers are encouraged by extension officers to eliminate these small fish. Meanwhile, farmers struggle to maintain strict tilapia growth levels in a monoculture system for long periods, meaning that total yields and productivity remain critically low [31]. In essence, as tilapia species in much of Africa are indigenous, compared to Asia where they are exotic, farmers end up growing small tilapias and/or a mix of other species throughout the year. Most of these farmers intermittently harvest fish from their ponds throughout the production cycle, almost exclusively for household consumption [32], thereby not allowing the tilapias the possibility of growing to full size. Public health statistics, meanwhile, highlight the urgency of improving food and nutrition security in rural Zambia and the critical role that SIS can play in supplying multiple nutrients including minerals, vitamins, essential fatty acids, and protein [18,33].

Farmers balance the needs of harvesting fish for food and generating cash. Governments and development organizations favor the latter commercialization narrative, which fails to recognize that many smallholder farmers simply do not have the financial means to grow tilapia unabatedly for the six or more months required to produce large fish [14]. In turn, the failure of these systems to improve livelihoods is often blamed on the lack of infrastructure and inputs (i.e., seed and feed) [34]. While the lack of input supply chains is a definitive barrier in sub-Saharan Africa, many policy and development practitioners fail to see aquatic ponds as a potential bank of highly nutritious foods that make up one part of a larger food system operated by a farmer. The vast supply of fish from capture fisheries, which dwarfs that of farmed fish in the region, is rarely acknowledged by studies that look to assess the role of ponds in improving food and nutrition security, despite an obvious overlap of competing fish products on the markets (wild versus farmed tilapia), and people’s fish consumption choices and preferences. 

There are calls for greater recognition of smallholder pond polyculture as a technology to help reach nutrition and health goals in Zambia [35]. For example, having learned from Bangladesh and Cambodia, WorldFish, an international research organization, funded polyculture trials in the north of the country, with promising results [36]. However, no studies tested such approaches directly with Zambian smallholder farmers, and none collected panel data that traced the consumption of fish from all sources to see how such a technology may fit into people’s fish-sourcing strategies. 

We investigate whether a polyculture system with various SIS could increase the supply of fish and the frequency of consumption. The polyculture systems introduced in this study are intentionally designed to grow several self-recruiting species in one pond. The objective of this research is to establish the potential contribution of aquaculture, and polyculture production specifically, to address household micronutrient sufficiency through the improved seasonal availability of fish. This requires looking at aquaculture in terms of the nutrients it can provide as opposed to solely producing large fish for markets. In a nutshell, this can be summarized as a nutrition-sensitive approach to rural smallholder farming in Zambia [37,38]. In other words, this entails placing nutrition at the center of the system rather than focusing on quantities produced and monetary outcomes. This approach prioritizes the food and nutrition security of poor households in addition to the productivity of farming systems, thus looking at access to and diversity of foods to ensure that food and nutrition security is met. To get a better sense of fish consumption choices that households make, we assessed all sources of fish in the region, including capture fisheries and dried fish markets. Therefore, we placed aquaculture and capture fisheries together in one aquatic food system that is interconnected, with many different types of aquatic foods and temporal benefits [39,40]. 

## 2. Methods and Materials

### 2.1. Sampling and Site Selection

Key informant interviews with extension officers from the government’s Department of Fisheries (DoF) were used to select the study sites in Luwingu District in northern Zambia. The extension officers were primarily responsible for all aquaculture development projects in the province and helped guide the site selection process. The intervention group was made up of people who trialed the pond polyculture intervention (referred to as the PP group), whereas the two control groups included people who practiced conventional “monoculture” pond farming (referred to as the MP group), and people who had no ponds at all and only practiced terrestrial agriculture (referred to as the AG group). The PP and AG groups were selected from the same villages (Luena and Isansa). This area was selected because the residents were new to aquaculture and the researchers did not want to interfere with, or contradict, established fish farming systems in the region. The MP group was selected from a village (Fisonge) close to the district capital, Luwingu, 78 km away from the other two groups, where there were more established fish farmers (see Figure 1). All households were primarily agricultural households. We aimed to recruit 20 households in each group, using focus-group discussions with village authorities to request volunteers. We were only able to recruit 17 households for the MP group. A total of 57 households were selected for the study. 

### 2.2. Intervention: Polyculture Pond Farming and Nutrition Training

The main intervention included stocking self-recruiting species in polyculture ponds. The species were selected based on a screening process that relied on a literature review of commonly consumed fish species, their nutrient profiles, and any evidence of pond trials in the region (details of the screening process are given in the Appendix A). In brief, the fish species selected for the trial were chosen because (1) they were often found in farmers’ ponds, (2) they had a high nutrient composition in the edible parts, and (3) there was some, albeit limited, information on their suitability for production in earthen ponds.

Farmers in the region typically cultivate the indigenous tilapias, *Oreochromis macrochir* and *Coptodon rendalli* [30]. *O. macrochir* and three other species were stocked in the PP group’s ponds as part of the intervention: a small adult-sized tilapia (*Tilapia sparrmanii*), another small cichlid (*Pseudocrenilabrus philander*), and a small barb (*Barbus trimaculatus*, which has since been changed to *Enteromius trimaculatus*). The *T. sparrmanii* and *O. macrochir* were sourced from local farmers’ ponds, while the *P. philander* and *B. trimaculatus* were sourced from the surrounding water bodies with the help of local fishermen. The number and stocking densities of the fish species are provided in Table 1. The *O. macrochir* were stocked as juveniles, while the SIS were mostly adult fish. Due to high mortality rates during the handling of the *P. philander* and *B*. *trimaculatus*, their weight and length measurements were combined.

The PP intervention group received additional training on pond management and on how human nutrition is improved through the consumption of fish, particularly on the benefits of consuming small fish whole for children and pregnant or lactating women, especially in the first 1000 days of life. The pond management training focused on three key issues that contradict the advice given to farmers by DoF extension officers and development workers. Participants were encouraged to:Take fish from their ponds whenever they wanted to, rather than at the end of the growth cycle (promoting intermittent harvesting).Cultivate a diversity of species and not eliminate SIS (promoting polyculture).Use natural rather than formulated feeds since the aim did not require maximizing the growth of a single species in a pond (promoting natural feeding regimes).

The trial was planned from the beginning of September 2019 to the end of March 2020. This constituted the beginning of spring moving into summer when air temperatures begin to warm and farmers in the region typically prepare their ponds for the coming rains. By the end of November, an annual national fishing ban implemented by the government prohibits all capture fisheries activities for three months (December, January, and February). The fishing ban is enforced every year during the spawning season as part of the Zambian government’s attempt to manage fish stocks and is applicable to all fisheries in Zambia except for Lakes Tanganyika and Kariba [44]. The fishing ban allowed for an additional seasonal dimension to ascertain whether fish supplies decreased during the ban and whether ponds might act as a substitute source of fish. This period, which is typically when farmers wait for the rains and start sowing their fields, is the time when food stocks from the previous year’s harvest are depleted, also known as the “hunger season” [45]. This, too, provides an additional seasonal dimension to the analysis from a food availability and access perspective. 

### 2.3. Data Collection

#### 2.3.1. Primary Data: Demographic Information and Fish Food Diaries

We collected demographic data, including household size, the age of the household head, marital status, years of education, disposable income, and the number and age of all children. Participants were trained on how to use fish food diaries to record the consumption of fish (but not other types of food) for the whole household, including the source of fish, to allow for comparisons between aquaculture and capture fisheries. Participants noted every instance when they consumed fish, including the species and form (dried/smoked/fresh), as well as the weight of fish. Participants used several household items, for example, cups, bowls, handfuls, and buckets to determine the quantities of fish. We converted these units of measurement for each fish species into kilograms. These conversion units were used throughout the study. The quantity of fish provided by participants referred to the total weight of all fish cooked and consumed on the day and not the weight of the edible portions. To validate quantities and descriptions, enumerators visited every month from September 2019 to March 2020, making a total of seven visits to each participating household. On visiting the household, enumerators discussed each entry to ensure accuracy. During this process, qualitative data were collected on how fish was sourced, cooked, portioned, and consumed, to provide a holistic view of people’s consumption habits and patterns. 

#### 2.3.2. Secondary Data: Nutrient Composition of Fish Species and Recommended Nutrient Intake 

A data set compiled by Hohenheim University includes the nutrient profiles of 43 species that are commonly consumed in Zambia [46]. The study collected multiple samples of each species, mostly from the Lake Bangweulu area, including both the dried and fresh forms. Fish were divided into “small”, “medium”, and “large” categories, based on size and edible portion (whole or filleted). The data set includes nutrient composition data per 100 g of edible portion for calcium (Ca), potassium (K), magnesium (Mg), iron (Fe), zinc (Zn), selenium (Se), chromium (Cr), and copper (Cu), as well as riboflavin (B_2_), niacin (B_3_), folate (B_9_), Cobalamin (B_12_), crude protein and omega-3 fatty acids: eicosapentaenoic acid (EPA), docosahexaenoic acid (DHA), and α-linolenic acid (ALA). The authors determined that these nutrients and omega-3 fatty acids were commonly found in fish compared to other animal-source foods and their contribution toward growth and development in the first 1000 days of life was a key focus. 

We used the recommended nutrient intake (RNI) for adults and children, as stipulated by the World Health Organization (WHO) and the Food and Agriculture Organization of the United Nations (FAO) [47], as a measure of nutrient security. Data for the intake of potassium was taken from the National Academies of Sciences, Engineering, and Medicine [48]. An RNI is the daily suggested amount of nutrients in grams for healthy individuals in specific age and sex groups, expressed as a percentage of reaching the daily target. In this case, the RNI averages for females across five age groups were used (see Appendix A). The RNI values for omega-3 fatty acids were derived from an expert consultation report [49]. There is no consensus on the RNI of omega-3 fatty acids for children and the RNI for adults differ, depending on contexts [50]. We established the RNI for omega-3 fatty acids by using the average energy requirements of females in different age groups [51], and then calculated the percentage of the energy requirements for each age group, as stipulated by the expert consultation report [49]. 

### 2.4. Analysis of Longitudinal Fish Consumption and Individual Nutrient Intake 

The quantity of individual fish species consumed by a household on a given day is the key unit of analysis in this study. Quantitative data were analyzed on how much fish was consumed, which species were cooked, in what form, and from which source, over a period of six months. The average consumption of fish per capita, per household, per day, was calculated by adding all the quantities of fish together and dividing by the number of people in each household, as well as the total number of days in each month. 

Dried and fresh fish weights are not directly comparable, since consuming the equivalent weight of dried fish to wet fish requires more units of fish to be caught/purchased. We calculated the difference in moisture content of wet fish compared to dry fish for every species using the study by Nölle and colleagues [46]. In some cases, where data were missing, we used similar fish species based on size and genus as a substitute (see Appendix A). By doing so, we calculated a wet weight equivalent in kilograms to be able to better compare the consumption of species. Given the small sample size in each farmer group and the non-normal distribution of fish weights, any statistical methods to compare differences in total fish weights between groups did not prove useful. 

There was no need to use a wet weight equivalent regarding the RNI calculations since the study by Nölle and colleagues collected the nutrient compositions of species in both dry and wet forms, respectively. We used the nutrient composition profiles of each species per 100-gram (g) edible portion (dry and wet values) to calculate the nutritional content of the fish consumed so that we could compare the total nutritional contributions between the groups. We multiplied the nutrient composition (in grams, milligrams, and micrograms of different nutrients) by the quantity of fish (in kilogram) consumed in a household each day (see Appendix A for more detail). We then divided each nutrient by the number of people in the household, subtracting infants (0–1 years old) that were still breastfeeding. The quantity of fish among all household members was divided equally. 

We acknowledge that adults and children consume different portion sizes of fish; however, we were regrettably unable to achieve this level of nuance for each unit of fish consumed in our approximation, given the vast diversity and sizes of fish species that came in both fresh and dried form. The nutrient composition for 100-gram edible portions was calculated for whole fish, including those parts of the fish that may have been discarded or thrown away, meaning that the results should be read with caution since we did not establish exactly which parts of the fish were consumed by whom. For larger fish, we used the nutrient composition of fillets, as per the study by Nölle and colleagues, when in fact some people in a household may have been eating different parts of a larger fish (i.e., head or tail). We only know the total quantity of fish consumed by a household and not the size of the individual units of fish consumed by each person. Where possible, we used qualitative interviews to determine whether certain species were likely to be consumed as adults or juveniles and either whole or filleted, and then used the corresponding nutrient values from the study by Nölle and colleagues (see Appendix A for more detail). Based on these data, we present the quantity of fish consumed on a given day and the contribution of this portion to meeting daily nutrient recommendations for each age group. This is calculated as a percentage of the daily RNI of all the nutrients assessed in this study for each age group and is then averaged for the household. 

We compared the quantities of fish consumed, the species, and the source between the three groups over time. We also compared the average amount, i.e., portion, of fish (for each species) per capita per day; by doing so, we can compare the contribution these fish made to the RNI of various nutrients, expressed as daily averages for the study period.

## 3. Results

The trial started on 9 September 2019 and ended on 31 March 2020, lasting for a total of 209 days. By November, one person from the PP group and one from the MP group had dropped out of the experiment. By January, two more people had dropped out of the AG group for undisclosed reasons. All subsequent analyses are based on the sample size of 53 households that provided complete data. 

Households from the PP and AG groups were from the same area and shared similar characteristics, although the MP group members were slightly older and wealthier on average, while the AG group members were notably younger and with smaller households (see Table 2). The PP and AG groups were located further down the escarpment, closer to Lake Bangweulu (see Figure 1). The Luena River flows through the area where the AG and PP groups were located and provides a local wetland fishery for these two groups. The MP group was slightly wealthier on average and was located further away, closer to markets and trade routes.

Each household consumed on average 40.6 kilograms (kg) of fish over 6 months. When considering the wet weight equivalent of fish, this resulted in 69.44 kg of fish on average or 0.33 kg of fish per household per day. With a total of 332 people in 53 households, this means a total of 11.1 kg of fish was available per person in each household over this period, resulting in just over 1.8 kg of fish per person per month and around 0.05 kg of fish per person per day. In total, all three groups consumed roughly the same amount of fish: the AG group consumed the total wet weight equivalent of 1243 kg of fish; the PP group consumed 1247 kg, while the MP group consumed 1191 kg. When dividing the quantity of fish by the number of people in the households, the AG group consumed 12.43 kg of fish per capita over 6 months, the PP group consumed 10.66 kg, and the MP group consumed 10.36 kg. The AG group had smaller household sizes on average. The average and ±standard deviation portion size of wet-weight-equivalent fish for a household on any given day was around 1.2 kg ± 1.68, which was portioned between 6.3 people on average, resulting in an average portion per person of 0.19 kg of fish per day. This was around 1 kg ± 1.6 for AG households, compared to 1.06 kg ± 1.6 for PP households, and 1.74 kg ± 1.8 for MP households. 

Figure 2 shows the average fish (wet weight equivalent in kilograms) per capita per day for each month, disaggregated by group. There was a general rise in the daily per capita average from September to November (note that the trial did not start on 1 September). The increase was sharpest for the AG and PP groups, who exponentially increased their consumption of fish just before the national fishing ban started in December. Coincidentally, there was a gradual decrease in fish consumption during the latter period, with the sharpest decrease reported by the AG group. The PP group started to harvest more fish from their ponds during this period. The MP group maintained a steadier per capita average of fish per day throughout the whole study period.

A total of 21 species were consumed across all households. Since some species were consumed less frequently than others, they were combined into a single species based on family and genus (see Appendix A for more detail). We categorized all these species into the 15 most frequently consumed species (see Table 3). 

The most frequently consumed fish were catfishes (*Clarias* spp.), as well as the smaller *T. sparrmanii* and the larger, and frequently cultivated, *C. rendalli*. These latter two tilapias were the most consumed fish in terms of total weight. However, as many of the small species were consumed dried, the wet weight equivalent of these fish far exceeded the total weight of *Clarias* spp. This means that a greater quantity of these small fish species was actually produced and consumed. 

This is better represented in Figure 3, which shows the same average quantity of wet weight equivalent (kg) fish per capita per day, disaggregated by group and source. The total weight of fish consumed and not the weight of the edible portions is given, although small fish were generally consumed whole. The PP and MP group members sourced between 10 to 20 g of fresh fish per capita per day from their ponds. The AG group members, who did not have ponds, sourced roughly double that from capture fisheries, and many of the species were the same as the ones found in the ponds of the PP and MP groups. 

The MP group hardly caught any fish from capture fisheries, compared to the other two groups; however, they did purchase a large quantity of dried fish from the market that were originally caught in capture fisheries located further away. Discussions with farmers revealed that these species were more available in the markets closer to the MP group, compared to the markets closer to AG and PP groups. Half of the fish consumed across all groups was either dried or smoked, especially fish purchased from local markets. In total, 1288.5 kg of fish was consumed fresh, whereas 863.7 kg was consumed dried and/or smoked, and the wet weight equivalent of the latter was far greater than that of fresh fish (2391.7 kg). Most of the fish (60%) was purchased, although there was a notable decrease in purchased fish from December onward, coinciding with the national fishing ban, meaning that households had to find alternative sources of fish. 

This decrease in fish consumption during the fishing ban was not as large for members of the MP group as it was for the AG and PP groups. The MP group started sourcing pelagic small fish and *L. stappersii* (Buka-Buka—a medium-sized perch) from capture fisheries further away; namely, from Lake Tanganyika, which was unaffected by the national fishing ban. According to interviews with farmers, despite the ban applied to Lake Bangweulu, where *Potamothrissa acutirostris*/*Poecilothrissa moeruensis* (chisense) is common, much of this fish was dried and stockpiled in November and illegally traded throughout the fishing-ban months. This fish was caught in the deeper pelagic zones on the western shore of the lake and landed in Samfya, meaning that it was processed in Luapula Province and then traded via road. When asked from which specific markets or vendors fish was accessed from, it was evident that the MP group had greater access to chisense and other pelagic small fish species as they were located along the main road by Luwingu, where fish was more frequently traded and sold (see Figure 1). 

During the fishing ban period, both the MP and PP groups increased the quantity of fish that they harvested from ponds. This gave these households a small additional source of fish during the closed fishing season. The PP group only started sourcing fish from their ponds in greater quantities once the fisheries were closed since the same species were readily available from capture fisheries in the open fishing season. During the closed fishing season, there was an increase in catfishes sourced from capture fisheries. When discussing the location whence fish was sourced, farmers stated that catfishes were widespread and were commonly found in rivers, streams, and ponds that were not usually monitored by DoF extension officers during the national fishing ban. 

Figure 4 provides more information on the quantity of fish consumed throughout the study period and how this varied between species and the three groups. The tilapia, *C. rendalli*, is the most consumed fish species (wet weight equivalent: kg/capita/day), and the MP group sourced almost a third of this from ponds. While this is one of the most widely cultivated fishes in the region, most of this fish was sourced from capture fisheries. The AG group consumed a larger quantity of *P. philander*, *T. sparrmanii* (two small cichlids), and *B. trimaculatus* (a small barb) than the PP group, despite these species being chosen for the polyculture intervention. The AG group consumed no *O. macrochir*, in contrast to the other two groups, as this was largely a cultivated tilapia species. 

The species of fish have varying nutrient compositions per 100 g edible portion (see Appendix A). This varies depending on the type of fish: for example, differences in fat content or micronutrients, whether the fish were consumed whole (including viscera and bones), or whether they were dried/smoked, all of which affect nutrient content. The catfishes and large tilapias are often consumed fresh after cooking. The small cichlids, such as *T. sparrmanii* and *P. philander*, if self-caught from capture fisheries were consumed fresh, although most are caught in large quantities and processed for sale in markets. Other small fish, such as *M. macrolepidotus*, *L. miodon*, and *P. acutirostris,* were almost exclusively consumed dried. Compared to larger fish, these smaller fish were consumed whole, including the viscera and bones. This is evident, for example, in the low amount of calcium provided by catfishes compared to the pelagic small fish species, because the latter were consumed whole with the bones (see Figure 5). Catfishes and larger cichlids, meanwhile, played an integral part in providing protein, mainly because of the size of the fillets that were consumed. The pelagic small fish species, such as *L. miodon* and *P. acutirostris*, provided far more omega-3 fatty acids per 100 g than the larger catfishes and tilapias. The smaller cichlids, such as *P. philander*, contributed the most omega-3 fatty acids, not because they have a particularly high concentration of fats but because of how much (total weight) was consumed. These small cichlids played an important role in contributing to the average RNI of calcium, riboflavin, and zinc, whereas catfishes provided fewer micronutrients despite being one of the most consumed fish species. Other notable fish species (*M. macrolepidouts* and *B. trimaculatus)*, although consumed in smaller quantities than the cichlids, still contributed high amounts of nutrients. 

Figure 6 shows the percentage of the RNI reached for each nutrient over time as an average per person per day for each study group. Overall, the entire sample achieved a daily average of 34.6% of their recommended protein intake, 8.6% of their recommended omega-3 fatty acids intake, and 48.2% of their recommended calcium intake. Participants in the study achieved almost double the daily recommended intake for vitamin B12 and selenium, on average. Since fish is known to contain high concentrations of these micronutrients, it is common for people to overreach the daily recommendation [52]. Over time, during the study period, the percentage of RNI achieved for most nutrients decreased, with the AG and PP groups experiencing the largest decreases from December onward. The MP group managed to avoid such a decrease, especially in their intake of omega-3 fatty acids. This was because of the high contribution of the pelagic fish, purchased from stocks caught from capture fisheries that are located further away, and because of the overall quantity consumed by the MP group (see Figure 3 and Figure 4). 

## 4. Discussion

The total quantity of fish consumed per capita over the whole study period was relatively the same for the three groups, pointing to people’s ability to find ways to satisfy their protein needs, in this case in the form of fish. The MP group managed to maintain a more consistent consumption of fish compared to the decreasing consumption experienced by the AG and PP groups. As a result of the national fishing ban, the latter groups almost doubled their fish consumption per capita in November in preparation for the inevitable decline in fish supplies starting in December, or for other unknown reasons to do with food availability during this time. This may be seen as indirect proof that fisheries management strategies are, indeed, successful in decreasing fishing activities and supplies. In anticipation of the ban, however, fishing pressure seems to increase in November, thus affecting the net impact that the ban may have on fish stocks. This study did not aim to assess the impact of the national fishing ban and other causes for this decline should be considered, such as the reduced catch per unit effort, resulting from an increase in rain and water levels making it difficult to access fishing grounds, especially in wetland swamps. Regardless, there was a clear trend of decreasing fish supplies experienced by all groups during this time, which is also regarded as the beginning of the “hunger season” for many poor and vulnerable Zambian families [45]. Such a drop in fish supplies, a primary animal-source food in this area, could exacerbate food and nutrition insecurity.

There is very little reliable information on the total fish yields in Zambian capture fisheries. Little is known about whether fisheries management strategies are successful; although, in general, there seems to be evidence of declining fish supplies from capture fisheries [53]. While the data in our study show a decline in the quantity and number of fish species from December onward, the MP group managed to shift their consumption of fish to dried pelagic species from other freshwater capture fisheries outside of Zambia, which were unaffected by the national fishing ban. Much of this fish is sourced from Malawi or Tanzania [54]. Such fish trade corridors along main roads allowed the MP group to access these fish species and, thus, maintain a higher intake of key micronutrients and omega-3 fatty acids. The MP households were made up of established fish farmers and are likely to be generally wealthier than non-fish farmers [55], another reason why this group could afford to purchase fish from markets more regularly. Many of the pelagic small fish species were not commonly traded in the AG and PP groups’ villages, given the poor condition of the roads; thus highlighting the importance of the accessibility of fish products. 

All three groups experienced a dip in protein intake over time, owing to a decrease in fish supplies; however, the omega-3 intake was variable between the three groups, owing to differences in species consumption. The pelagic small fish species contained high amounts of fatty acids, and they were consumed whole including the viscera and bones. This points to the importance of these species and capture fisheries in providing access to key nutrients. While these fishes may not be available in certain areas, other small fishes, if consumed whole and in sufficient quantities, can also be a critical source of omega-3 fatty acids. The small cichlids *T. sparrmanii* and *P. philander* contributed much of the omega-3 fatty acids for the AG and PP groups, suggesting that they may be good candidate species for polyculture systems. It is important to consider the nutrient composition of edible portions, as well as the total quantity consumed. While some fish species may have exceptionally high concentrations of certain micronutrients and fatty acids, they may be consumed less frequently. This points to the importance of assessing not only edible portions correctly but also the total quantity and frequency of fish species consumed. 

A large quantity of fish was consumed by these households (over 11 kg of fish per capita during a six-month period). Fish was consumed almost every second day. This is above the annual average for Africa (10.8 kg/capita/year) and far above the annual average of East Africa (4.8 kg/capita/year) but below the annual average for West Africa (15.3 kg/capita/year) [9]. Considering that we measured this consumption for half a year and during the time of the national fishing ban, we can assume that people in our study consumed higher amounts of fish on an annual basis. It is worth mentioning that this study did not evaluate other animal-source foods that households consumed, nor did we assess whole diets—for example, how much, in terms of cereals, dark green leafy vegetables, and fruits, was consumed. It is, therefore, unclear what other foods people consumed during this time; however, there is evidence that people in this region have little access to other animal-source foods and that fish is the primary protein source throughout the year [16]. 

The primary purpose of this research was not to establish which aquatic food system provided a better source of fish and nutrients, per se, but to establish whether polyculture fish farming can provide a significant and alternative source of fish. When looking specifically at the role of ponds in supplying fish, it was clear that they served a similar purpose for the MP and PP groups. The MP group claimed to grow tilapia for markets by operating strict monoculture systems for several months; however, most of these farmers harvested fish from their ponds sporadically throughout this period. This group even harvested *P. philander* and *T. sparrmanii* from their ponds (two fish that were selected for the polyculture intervention), suggesting that some, if not most, farmers in the region probably operate polyculture ponds by default. The fact that most small-scale ponds are, in fact, polyculture systems is rarely acknowledged in assessments of small-scale aquaculture in sub-Saharan Africa.

The PP group consumed a slightly larger quantity of fish from ponds than the MP group did, which was important from a food and nutrition security perspective, as they did not have the same access to fish markets as the latter group. The PP group, then, had an additional source of fish that the AG group did not have. The ponds provided an important source of fish, particularly during the months of the national fishing ban when both the PP and MP groups increased their consumption from ponds. The PP group tended to harvest less fish from capture fisheries during this time, as fish was available from their ponds. Polyculture ponds that can provide fish all year round, but especially during the national fishing ban, may be beneficial for fisheries management as well as food security objectives. It is also likely that the PP group spent less money on buying fish from markets as they had access to fish from their ponds. The PP group sourced notably less fish from markets than the other two groups. Therefore, ponds can provide additional fish, but low yields from ponds mean that they cannot substitute fish from capture fisheries. 

Since the PP and MP groups harvested fish from their ponds for consumption, it stands to reason that polyculture may provide two production strategies for farmers: (1) they can use ponds exclusively, and almost daily, as a source of diverse fish for human consumption; or (2) they can integrate polyculture with the aim of additionally producing larger fish (tilapia) for markets, since there may be niche opportunities for growing both tilapia and SIS at the same time [27]. Though the biophysical aspects of the latter were not tested in this study, some farmers from the sample expressed their interest in operating ponds with diverse fish species for household consumption whilst at the same time operating ponds with single species strictly for sale. Other farmers saw an opportunity to do both at the same time in one pond. The intentional recruitment of SIS species into ponds can be a sound livelihood activity for semi-controlled pond systems, as they are in Bangladesh [28]. The value of polyculture ponds is to provide more fish and a diversity of fish species—small and large—for consumption and for sale, and to extend the season of consumption, minimizing the reliance on capture fisheries and the negative effect of the fishing ban.

An extensive, low-input system with multiple highly nutritious fish species enables not only management techniques, such as phytoplankton-based or periphyton-based growth, but also allows for partial harvesting throughout the production cycle. This may be more complementary to the conditions and characteristics of smallholder aquaculture in sub-Saharan Africa. A high diversity of fish species, the inclusion of indigenous species, and polyculture production methods are likely to be more compatible with smallholder aquaculture at this stage of aquaculture development on the subcontinent. This is especially the case for poorer farmers who struggle to produce for markets and in areas where malnutrition and food and nutrition security are major development challenges. The potential to widen the parameters for diverse species selection must be considered, to allow for the growth and development of aquaculture in the region. 

## 5. Conclusions

By using a food systems lens in assessing the contribution of various aquatic systems, we were able to ascertain a more complete picture of how households in rural Zambia consumed fish. We achieved this by looking at all aquatic food systems in the region and placing human nutrition at the center of the analysis. We considered, specifically, the individual species produced in various systems with the goal of improving access to these species. The study took place during seasonal shifts, including weather changes as well as fisheries management interventions and food scarcity fluctuations, which helped to better understand fish consumption trends. 

This research provided evidence that people’s ability to shift their sourcing strategies of fish, due to various circumstances, was the most important factor in meeting their overall nutritional needs. A diversity of fish species, a diversity of sources, and the ability to adapt and change sourcing (and expenditure) strategies provided households with a more flexible pathway to food and nutrition security. 

Polyculture ponds can play a complementary role to the current tilapia production paradigms implemented in Zambia and other sub-Saharan countries, which tend to focus on the productivity of tilapia under supposedly monoculture systems. Aquaculture development must be positioned within the larger aquatic resource system. This should encompass assessing the contribution of diverse fish species from a vast array of different inland water bodies, including lakes and rivers, especially because pelagic small fish species contributed significantly to micro-nutrient and fatty acid intake compared to other species in this study. Development projects should continue to develop the infrastructure and supply chains associated with the tilapia industry in Zambia so that more small-scale farmers can participate successfully (see 55). Some farmers may opt for more intensive and commercial forms of aquaculture that rely on the monoculture production of individual species; however, farmers who are unable to consistently produce single species to commercial sizes could adopt polyculture pond farming as a potential solution, to better utilize water resources on the farm and maximize nutrient yield. 

The best way to assess the efficacy of a food system is to assess how well it provides nutritious foods in comparison to other, similar systems in the area. This further provides a strong justification to continue placing aquaculture and capture fisheries in an interconnected continuum, rather than as separate systems, with a focus on the diversity of species and systems [56]. Nutrition-sensitive approaches must avoid the same trap of “productionist” approaches that only look at the potential of a single system or single food, without considering complementary or competing systems. Assessing these systems is not only about the bioavailability or economic accessibility of diverse foods but also about the choices and strategies that people make, based on varying contexts and drivers that differ from season to season. While the polyculture pond approach aims to improve access to a diversity of fish species, thereby improving dietary diversity and nutrition and health outcomes, there are dimensions of the approach that require further investigation to properly assess how nutrition-sensitive these systems truly are. Namely, this means assessing the potential income of these systems and also whether the approach empowers and improves women’s access to and control over resources, ultimately lifting their social status [38]. While the latter was not the focus of this research, studies in Bangladesh suggest that backyard-style pond farming has been beneficial for women’s empowerment [23]. Coupled with the potential of integrating aquaculture with agricultural activities on smallholder farms, the pond polyculture system can have a positive impact on livelihoods as well as food and nutrition security. 

## Figures and Tables

**Figure 1 foods-11-01334-f001:**
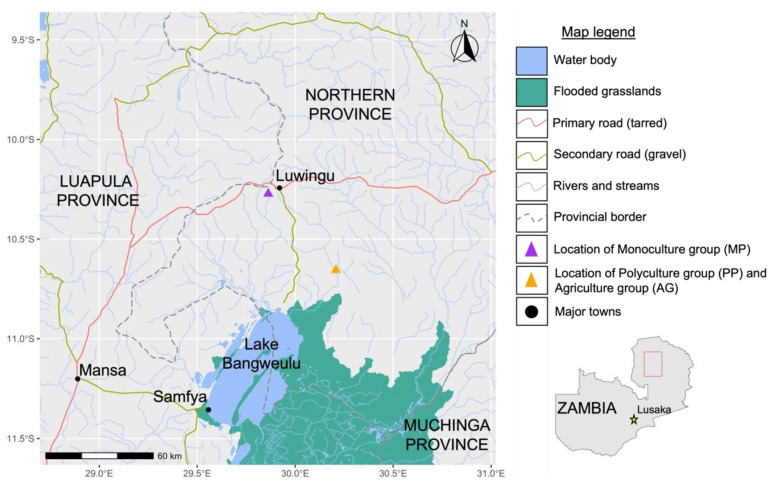
Map of study site locations in Northern Province, Zambia. Data for the flooded grasslands biome is from Terrestrial Ecoregions of the World (TEOW) [41]. Rivers and water bodies are from the HydroATLAS [42]) and HydroATLAS-Zambia [43].

**Figure 2 foods-11-01334-f002:**
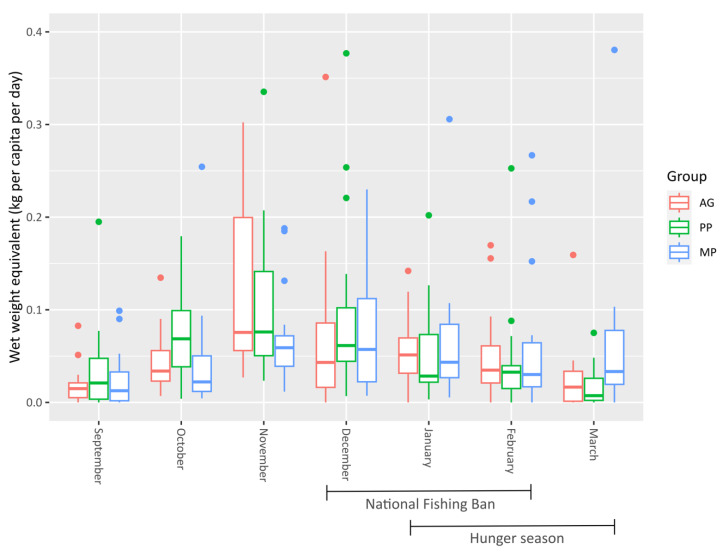
The monthly quantity of fish consumed, with the wet weight equivalent in kilograms per capita per day, for the three treatment groups. Outliers above 0.4 kg have been truncated for clarity, removing 4 observations.

**Figure 3 foods-11-01334-f003:**
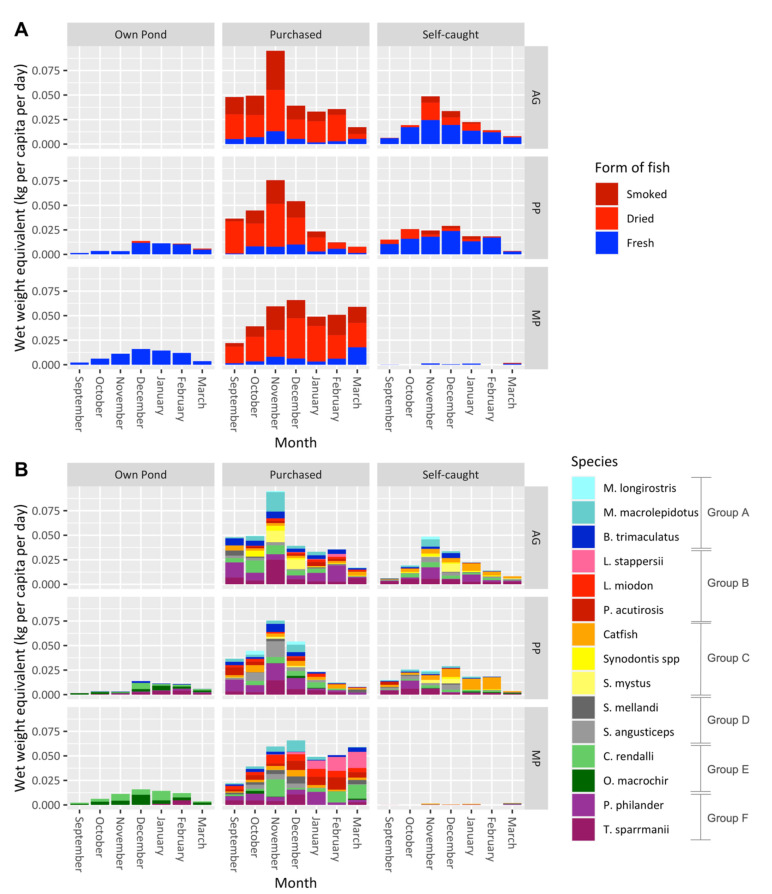
Monthly quantity of fish consumed as a wet weight equivalent (kg/capita/day) according to the three study groups and sources of fish: (**A**) form of preparation of fish; (**B**) species. Group A: mormyrids and local barbs, generally consumed as juveniles and caught in small lagoons and channels in wetlands. Group B: caught in the pelagic zones of large, further-away fisheries and frequently traded throughout Zambia. Group C: catfishes of all sizes and some of the most frequently consumed fish in the region. Group D: large, robust cichlids caught in nets or with hand-lines. Group E: widely consumed tilapias that are frequently cultured in ponds but are mainly sourced from capture fisheries. Group F: small, wild cichlids that are widely consumed and usually gain entry into farmers’ ponds.

**Figure 4 foods-11-01334-f004:**
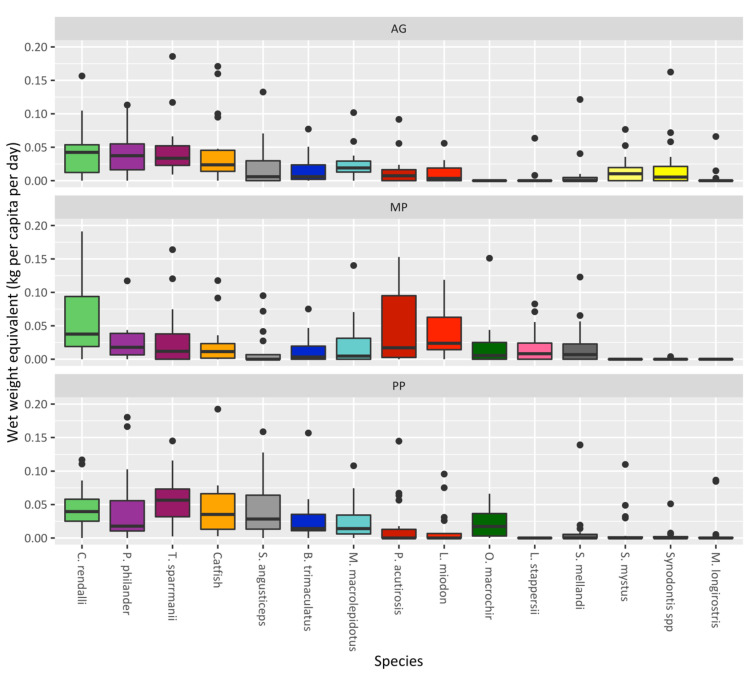
The average quantity of fish consumed (wet weight equivalent: kg species/capita/day), disaggregated by the three study groups. From left to right: the species are ordered as the most to least consumed fish on average for the whole sample of households over the entire study period, in terms of the total wet weight equivalent (kg). Outliers above 0.2 kg/capita/day have been truncated for clarity, thus removing 18 observations.

**Figure 5 foods-11-01334-f005:**
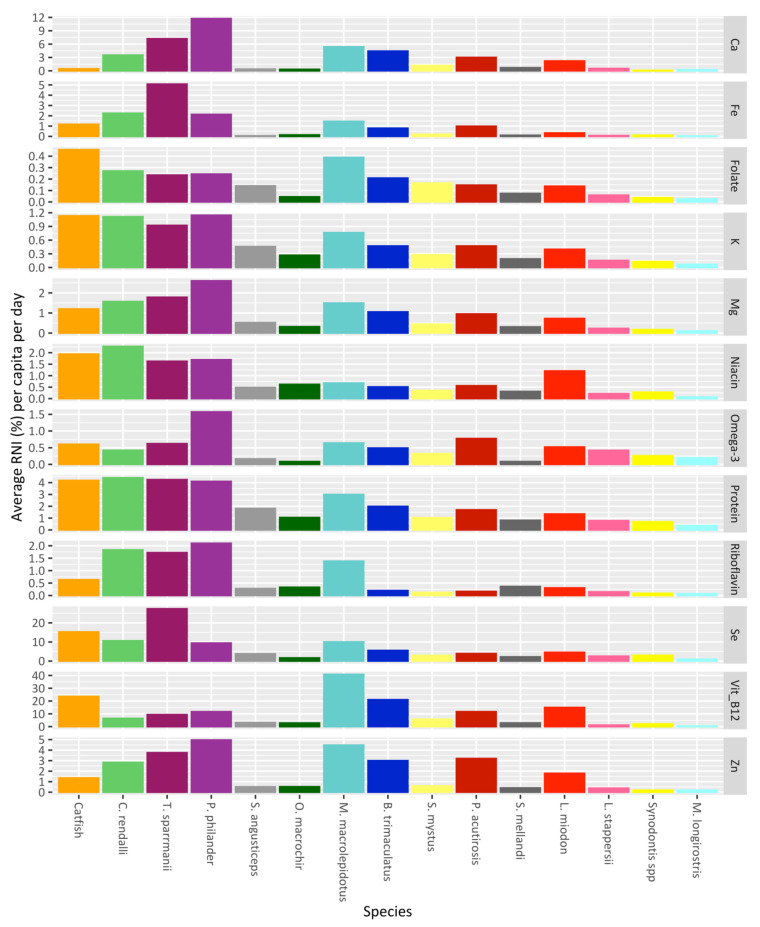
Average consumption of each nutrient per species as a percentage of the recommended nutrient intake (RNI) achieved for each nutrient per capita per day. From left to right, the species are ordered as the most to least consumed fish for the whole sample of households over the entire study period, in terms of the total weight (kg) of fish (i.e., not the wet weight equivalent).

**Figure 6 foods-11-01334-f006:**
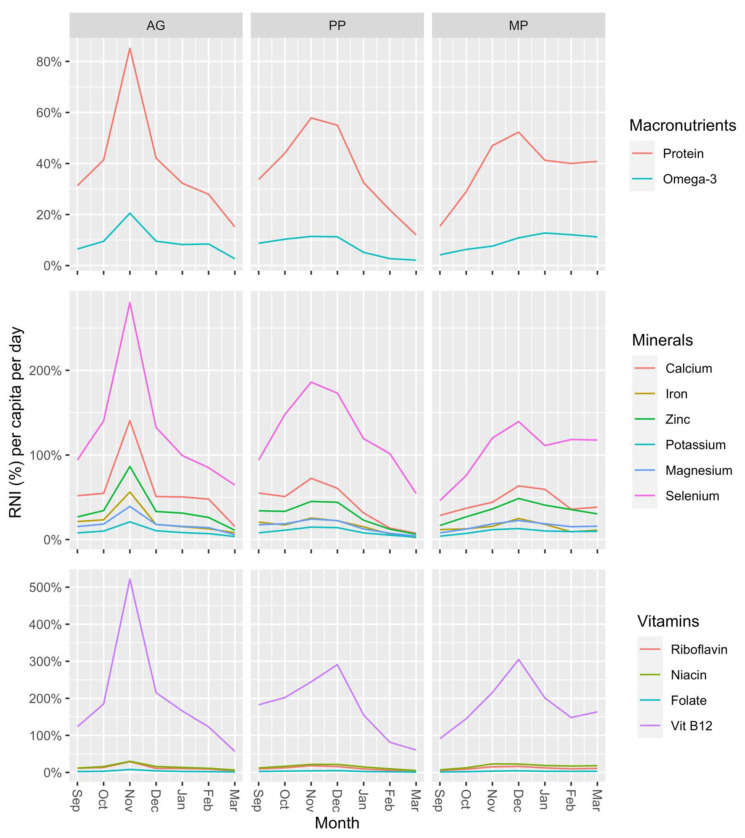
The percentage of the recommended nutrient intake (RNI) reached per capita per day of selected nutrients: minerals, vitamins, and omega-3 fatty acids derived from the consumption of fish over 6 months (September 2019–March 2020), disaggregated according to the three study groups.

**Table 1 foods-11-01334-t001:** Stocking data for polyculture trial including one commercial tilapia species (*O. macrochir*) and three Small Indigenous Species (SIS).

Species	Total Fish	Number of Fish Stocked in Ponds (*n* = 20)		StockingDensity (Fish/m^2^)		Weight of Fish (g)		Length of Fish (cm)	
		M	SD	M	SD	M	SD	M	SD
*O. macrochir*	8554	427.26	11.63	1.98	1.01	13.37	4.63	8.56	1.00
**Small Indigenous Species**									
*T. sparrmanii*	2000	100	0.00	0.46	0.24	5.04	1.01	5.92	0.60
*P. philander*	2000	100	0.00	0.46	0.24	1.80 †	0.84 †	4.10†	0.75 †
*B. trimaculatus*	1000	50	0.00	0.23	0.12				

† *P. philander* and *B. trimaculatus* were combined at the time of stocking; weights and lengths reflect a random sampling of the species mix.

**Table 2 foods-11-01334-t002:** Household descriptive statistics.

	Total (*N* = 53) ^a^	Polyculture (PP) (*n* = 19)	Monoculture (MP) (*n* = 16)	Agriculture (AG) (*n* = 18)
Age (Mean Years ± SD)	40.6 ± 11.4	39.9 ± 10.1	44.9 ± 12.2	37.4 ± 11.5
Education (Mean Years ± SD)	7.6 ± 2.0	6.7 ± 2.3	7.7 ± 1.9	8.5 ± 1.3
Household size (Mean No. of People ± SD)	6.3 ± 2.5	6.2 ± 2.6	7.2 ± 2.4	5.6 ± 2.5
Number of Children (Mean No. ± SD)	4.3 ± 2.4	4.3 ± 2.4	4.7 ± 2.6	3.9 ± 2.4
Marital Status (Freq. and % Single)	14 (26%)	6 (32%)	1 (6%)	6 (33%)
Head of Household (Freq. and % Female-headed)	13 (25%)	6 (32%) ^b^	1 (6%)	6 (33%)
Average Disposable Income (Mean ZMW ^c^ ± SD)	5265 ± 7982	5237 ± 10,943	6215 ± 6200	4449 ± 5709

All values are mean and standard deviations unless otherwise specified. ^a^ The original sample was *N* = 57 but four participants dropped out of the experiment. ^b^ Only one woman was married as well as being the head of the household. All single women were heads of the household. ^c^ ZMW = Zambian Kwacha.

**Table 3 foods-11-01334-t003:** Categories and names of the fish species consumed, including total frequency (number of times consumed) and total quantity (kilograms consumed), represented as the measured weight and wet weight equivalents.

Category *	Scientific Name	Local Name	Frequency	Measured Weight (kg)	Wet Weight Equivalent (kg)
A: Mormyrids and barbs (wetland species)	*Mormyrus longirstris*	Mbubu	38	13.7	33.9
	*Marcusenius macrolepidotus*	Mintesa	278	119.6	234.1
	*Barbus trimaculatus* †	Mushipa	243	122	242.5
B: Pelagic small/medium fish	*Luciolates stappersii*	Buka-Buka	59	63.3	141.8
	*Limnothrissa miodon* and *Stolothrissa tanganicae*	Kapenta	138	71.9	197.6
	*Potamothrissa acutirostris* and *Poecilothrissa moeruensis*	Chisense	133	66.3	214.1
C: Catfishes (large and small)	*Clarias* spp.	Milonge	465	333.4	350.7
	*Syndontis* spp.	Cingongo	79	44.9	70.3
	*Schilbe mystus*	Lupata	41	70.7	120
D: Large cichlids	*Sargochromis mellandi*	Imbelya	89	75.1	139.8
	*Serranochromis angusticeps*	Polwe	133	157.7	274.4
E: Tilapias (often cultivated)	*Coptodon rendalli*	Mpende	326	388	508.4
	*Oreochromis machrochir*	Nkamba	121	178	193.3
F: Small cichlids from local capture fisheries	*Pseudocrenilabrus philander*	Cikundu	384	165.2	480.2
	*Tilapia sparrmanii*	Matuku	553	282.3	479.1

* Letters A–F in the Category column correspond to fish groups in Figure 3. † *Barbus trimaculatus* has since been changed to *Enteromius trimaculatus*.

## Data Availability

The data is available in a publicly accessible repository [Harvard Dataverse, Pond Polyculture Approaches in Luwingu, Zambia] at https://doi.org/10.7910/DVN/EKFJYE.

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
