# Peer review of "The Role of Aquaculture and Capture Fisheries in Meeting Food and Nutrition Security: Testing a Nutrition-Sensitive Pond Polyculture Intervention in Rural Zambia"

_foods, 2022, doi:10.3390/foods11091334_

Round 1

Reviewer 1 Report

A wonderfully detailed longitudinal intervention study of polyculture systems using two controls (monoculture, agriculture).  The only limitation is the sample size but that isn't a reason not to publish. A nuanced look at a multi-dimensional issue. I particularly liked Figures 3 and 5.  

Introduction

Consider starting at new paragraph with the sentence “We investigate whether polyculture systems…”

Methods

When were fish stocked in ponds? Were the fish of harvestable size when the study began or was there a ramp-up needed? Was this taken into account in the intake data?

Results

Can you explain more about the national fishing ban? Is this an annual event or a on-time event related to the pandemic? Why was the ban instituted?

It would be of interest to know if own pond fish replaced or were in addition to purchased or self-caught.  Put another way, is this an issue of substitution or addition?

Discussion

“The fact that most small-scale ponds are, in fact, polyculture systems is rarely acknowledged in assessments of small-scale aquaculture in sub Saharan Africa.”

This is a fascinating finding.

Conclusion

Replace “couple” with “coupled”

Author Response

We sincerely thank the reviewer for the positive comments and are grateful for the time taken to engage with this piece. Thank you.

We have made all structural and grammatical changes the reviewer suggested.

In terms of the query regarding when fish were stocked, the size of fish on stocking, and whether this was taken into account in the nutrient data, our response is as follows: In the methods we have provided a little more detail as to the stocking of fish though we point the reviewer towards Table 1 (Page 5), which details the mean weight and length of all the fish stocked. We have now added a sentence stating that the tilapia (O. macrochir) were intentionally stocked as juveniles while the SIS were stocked mostly as adults. Thank you for this suggestion. Further down in the methods on Page 8, we made more clear the limitation that we were not always able to account for the sizes of fish or whether fish were consumed whole/filleted by each member of the family on each day. The nutrient data from Hohenheim University that we relied on had nutrient values for fish either filleted or whole. We used qualitative interviews to determine how fish were mostly consumed, meaning that for larger fish we used nutrient values for fillets, while for smaller fish we used nutrient values for whole fish. Regrettably, we didn’t always know whether a tilapia was consumed large or small, for example. This is made clearer now in the methods, thank you.

The other request for more detail on the fishing ban has been provided on Page 6. The fishing ban is an annual occurrence aimed to restrict fishing activities during the spawning season. Thank you for this suggestion.

We hope that these revisions meet your standards and we thank you again for your review.

Reviewer 2 Report

The main objective of this research was not to establish which aquatic food system provided a better source of fish and nutrients. The aim was to establish whether polyculture aquaculture can provide a significant alternative source of fish. The role of aquaculture and capture fisheries in achieving food and nutrition security tests the implementation of nutrition-sensitive pond polyculture in rural areas such as Zambia. It is a specific job, in other countries it is a proven technology and the conversion of farmers to aquaculture has grown statistically.
 It has been shown in other studies that there are no nutritionally poor species and small endemic species contain high values ​​of protein and fatty acids of importance for public health, in their different stages of the crop cycle.
It is desirable for the authors to strengthen the importance of contributing, through this type of production system, quality products and safety as they are semi-controlled systems.
Biosecurity is provided during cultivation.
It is necessary to highlight that drying and smoking technologies are conservation alternatives that, in the international market, raise the economic and nutritional value of fishery and aquaculture products. Not only eating fresh and high weight products is desirable in food safety.

Author Response

We thank the reviewer for their time and expertise on reviewing this paper. We are grateful for these comments and suggestions.

Indeed, the Zambian polyculture system differs to more established polyculture systems in Asia, for example. We have improved our discussion on this in the introduction (page 2). Thank you for this suggestion. The difference is that most Zambian pond systems are unintentionally and by default, polyculture systems. We agree also that these are semi-controlled systems and we provide more depth as to how these Zambian systems ought to be operated in the discussion (page 18). Thank you for raising this.

Although we agree that there are ‘no nutritionally poor species’ the range in micronutrients and potential contribution to overall diets is likely to be very different between species. So we agree with the reviewer that all fish provide good amounts of protein, fatty acids and other micronutrients, however, we purposively stocked those fish that had superior nutrient profiles per 100g edible portions and that were also feasibly cultivable in farmers’ ponds. We detailed this in the methods. The purpose of this paper was to show the important contribution of highly nutritious fish and whether polyculture could add to or substitute these sources. As per the reviewer’s comments however, we have made it clear in the discussion that these polyculture systems provide additional fish and are not a substitute for capture fisheries. Thank you for raising this to our attention.

We are regrettably unable to make too many points as to the biophysical, and specifically the biosecurity, of these pond systems. The biophysical data from this experiment was not suitable for publication and was omitted from the paper entirely. We would be hesitant to make any further arguments on this. However, as to the Reviewer’s comments we do aim to show that while dried fish is the main form of fish, there is a benefit also (cultural and nutritional) to consuming fresh fish, and that since many farmers do not have access to fresh fish this can be a welcome, high-quality product that has not degraded. We do not completely agree that dried fish are always better for conservation as the equivalent weight of dried fish to wet fish requires significantly more individual units of fish. There are also tremendous food losses and wastage during the drying process. This is why we did not stray into this discussion as it is highly complex and was not the purpose of this paper -  it would take up too much room in the discussion. We do however highlight the benefit of using wet weight equivalents. We thank the reviewer for bringing this to our attention.

We trust that these revisions meet the reviewer’s standards, and we thank you again for taking the time to review this paper.

Round 2

Reviewer 2 Report

It is desirable to make a better control of the experimental design for the continuity or new study, as an alternative for the population to be able to generate food by this technology as the study already carried out is presented. The document can provide a sufficient percentage.